# Universal Option Models

**Hengshuai Yao, Csaba Szepesvári, Rich Sutton, Joseph Modayil**
Department of Computing Science
University of Alberta
Edmonton, AB, Canada, T6H 4M5
`hengshua,szepesva,sutton,jmodayil@cs.ualberta.ca`

**Shalabh Bhatnagar**
Department of Computer Science and Automation
Indian Institute of Science
Bangalore-560012, India
`shalabh@csa.iisc.ernet.in`

## Abstract

We consider the problem of learning models of options for real-time abstract planning, in the setting where reward functions can be specified at any time and their expected returns must be efficiently computed. We introduce a new model for an option that is independent of any reward function, called the *universal option model (UOM)*. We prove that the UOM of an option can construct a traditional option model given a reward function, and also supports efficient computation of the option-conditional return. We extend the UOM to linear function approximation, and we show the UOM gives the TD solution of option returns and the value function of a policy over options. We provide a stochastic approximation algorithm for incrementally learning UOMs from data and prove its consistency. We demonstrate our method in two domains. The first domain is a real-time strategy game, where the controller must select the best game unit to accomplish a dynamically-specified task. The second domain is article recommendation, where each user query defines a new reward function and an article's relevance is the expected return from following a policy that follows the citations between articles. Our experiments show that UOMs are substantially more efficient than previously known methods for evaluating option returns and policies over options.

## 1  Introduction

Conventional methods for real-time abstract planning over options in reinforcement learning require a single pre-specified reward function, and these methods are not efficient in settings with multiple reward functions that can be specified at any time. Multiple reward functions arise in several contexts. In inverse reinforcement learning and apprenticeship learning there is a set of reward functions from which a good reward function is extracted [Abbeel et al., 2010, Ng and Russell, 2000, Syed, 2010]. Some system designers iteratively refine their provided reward functions to obtain desired behavior, and will re-plan in each iteration. In real-time strategy games, several units on a team can share the same dynamics but have different time-varying capabilities, so selecting the best unit for a task requires knowledge of the expected performance for many units. Even article recommendation can be viewed as a multiple-reward planning problem, where each user query has an associated reward function and the relevance of an article is given by walking over the links between the articles [Page et al., 1998, Richardson and Domingos, 2002]. We propose to unify the study of such problems within the setting of real-time abstract planning, where a reward function can be speci-

fied at any time and the expected option-conditional return for a reward function must be efficiently computed.

*Abstract planning*, or *planning with temporal abstractions*, enables one to make abstract decisions that involve sequences of low level actions. Options are often used to specify action abstraction [Precup, 2000, Sorg and Singh, 2010, Sutton et al., 1999]. An option is a course of temporally extended actions, which starts execution at some states, and follows a policy in selecting actions until it terminates. When an option terminates, the agent can start executing another option. The traditional model of an option takes in a state and predicts the sum of the rewards in the course till termination, and the probability of terminating the option at any state. When the reward function is changed, abstract planning with the traditional option model has to start from scratch.

We introduce universal option models (UOM) as a solution to this problem. The *UOM* of an option has two parts. A *state prediction* part, as in the traditional option model, predicts the states where the option terminates. An *accumulation* part, new to the UOM, predicts the occupancies of all the states by the option after it starts execution. We also extend UOMs to linear function approximation, which scales to problems with a large state space. We show that the UOM outperforms existing methods in two domains.

## 2 Background

A finite Markov Decision Process (MDP) is defined by a discount factor $\gamma \in (0,1)$, the state set, $\mathbb{S}$, the action set, $\mathbb{A}$, the immediate rewards $\langle \mathcal{R}^a \rangle$, and transition probabilities $\langle \mathcal{P}^a \rangle$. We assume that the number of states and actions are both finite. We also assume the states are indexed by integers, *i.e.,* $\mathbb{S} = \{1, 2, \ldots, N\}$, where $N$ is the number of states. The immediate reward function $\mathcal{R}^a : \mathbb{S} \times \mathbb{S} \to \mathbb{R}$ for a given action $a \in \mathbb{A}$ and a pair of states $(s, s') \in \mathbb{S} \times \mathbb{S}$ gives the mean immediate reward underlying the transition from $s$ to $s'$ while using $a$. The transition probability function is a function $\mathcal{P}^a : \mathbb{S} \times \mathbb{S} \to [0, 1]$ and for $(s, s') \in \mathbb{S} \times \mathbb{S}$, $a \in \mathbb{A}$, $\mathcal{P}^a(s, s')$ gives the probability of arriving at state $s'$ given that action $a$ is executed at state $s$.

A (stationary, Markov) *policy* $\pi$ is defined as $\pi : \mathbb{S} \times \mathbb{A} \to [0, 1]$, where $\sum_{a \in \mathbb{A}} \pi(s, a) = 1$ for any $s \in \mathbb{S}$. The *value* of a state $s$ under a policy $\pi$ is defined as the expected return given that one starts executing $\pi$ from $s$:

$$V^\pi(s) = E_{s,\pi}\{r_1 + \gamma r_2 + \gamma^2 r_3 + \cdots\}.$$

Here $(r_1, r_2 \ldots)$ is a process with the following properties: $s_0 = s$ and for $k \geq 0$, $s_{k+1}$ is sampled from $\mathcal{P}^{a_k}(s_k, \cdot)$, where $a_k$ is the action selected by policy $\pi$ and $r_{k+1}$ is such that its conditional mean, given $s_k, a_k, s_{k+1}$ is $\mathcal{R}^{a_k}(s_k, s_{k+1})$. The definition works also in the case when at any time step $t$ the policy is allowed to take into account the history $s_0, a_1, r_1, s_1, a_2, r_2, \ldots, s_k$ in coming up with $a_k$. We will also assume that the conditional variance of $r_{k+1}$ given $s_k$, $a_k$ and $s_{k+1}$ is bounded.

The terminology, ideas and results in this section are based on the work of [Sutton et al., 1999] unless otherwise stated. An option, $o \equiv o\langle \pi, \beta \rangle$, has two components, a policy $\pi$, and a *continuation function* $\beta : \mathbb{S} \to [0, 1]$. The latter maps a state into the probability of continuing the option from the state. An option $o$ is executed as follows. At time step $k$, when visiting state $s_k$, the next action $a_k$ is selected according to $\pi(s_k, \cdot)$. The environment then transitions to the next state $s_{k+1}$, and a reward $r_{k+1}$ is observed.[1] The option terminates at the new state $s_{k+1}$ with probability $1 - \beta(s_{k+1})$. Otherwise it continues, a new action is chosen from the policy of the option, *etc.* When one option terminates, another option can start.

The *option model* for option $o$ helps with planning. Formally, the model of option $o$ is a pair $<R^o, p^o>$, where $R^o$ is the so-called *option return* and $p^o$ is the so-called *(discounted) terminal distribution* of option $o$. In particular, $R^o : \mathbb{S} \to \mathbb{R}$ is a mapping such that for any state $s$, $R^o(s)$ gives the total expected discounted return until the option terminates:

$$R^o(s) = \mathbb{E}_{s,o}[r_1 + \gamma r_2 + \cdots + \gamma^{T-1} r_T],$$

where $T$ is the random termination time of the option, assuming that the process $(s_0, r_1, s_1, r_2, \ldots)$ starts at time 0 at state $s_0 = s$ (*initiation*), and every time step the policy underlying $o$ is followed to get the reward and the next state until termination. The mapping $p^o : \mathbb{S} \times \mathbb{S} \to [0, \infty)$ is a function

that, for any given $s, s' \in \mathbb{S}$, gives the discounted probability of terminating at state $s'$ provided that the option is followed from the initial state $s$:

$$
\begin{aligned}
p^o(s, s') &= \mathbb{E}_{s,o}\big[\, \gamma^T \mathbb{I}_{\{s_T = s'\}} \,\big] \\
&= \sum_{k=1}^{\infty} \gamma^k \, \mathbb{P}_{s,o}\{s_T = s', T = k\}.
\end{aligned}
\tag{1}
$$

Here, $\mathbb{I}_{\{\cdot\}}$ is the indicator function, and $\mathbb{P}_{s,o}\{s_T = s', T = k\}$ is the probability of terminating the option at $s'$ after $k$ steps away from $s$.

A semi-MDP (SMDP) is like an MDP, except that it allows multi-step transitions between states. A MDP with a fixed set of options gives rise to an SMDP, because the execution of options lasts multiple time steps. Given a set of options $\mathcal{O}$, an *option policy* is then a mapping $h : \mathbb{S} \times \mathcal{O} \to [0, 1]$ such that $h(s, o)$ is the probability of selecting option $o$ at state $s$ (provided the previous option has terminated). We shall also call these policies *high-level* policies. Note that a high-level policy selects options which in turn select actions. Thus a high-level policy gives rise to a standard MDP policy (albeit one that needs to remember which option was selected the last time, *i.e.,* a history dependent policy). Let $\mathrm{flat}(h)$ denote the standard MDP policy of a high-level policy $h$. The value function underlying $h$ is defined as that of $\mathrm{flat}(h)$: $V^h(s) = V^{\mathrm{flat}(h)}(s), s \in \mathbb{S}$. The process of constructing $\mathrm{flat}(h)$ given $h$ and the options $\mathcal{O}$ is the flattening operation. The model of options is constructed in such a way that if we think of the option return as the immediate reward obtained when following the option and if we think of the terminal distribution as transition probabilities, then Bellman's equations will formally hold for the tuple $\langle \gamma = 1, \mathbb{S}, \mathcal{O}, \langle R^o \rangle, \langle p^o \rangle \rangle$.

## 3 Universal Option Models (UOMs)

In this section, we define the UOM for an option, and prove a universality theorem stating that the traditional model of an option can be constructed from the UOM and a reward vector of the option. The goal of UOMs is to make models of options that are independent of the reward function. We use the adjective "universal" because the option model becomes universal with respect to the rewards. In the case of MDPs, it is well known that the value function of a policy $\pi$ can be obtained from the so-called *discounted occupancy function* underlying $\pi$, e.g., see [Barto and Duff, 1994]. This technique has been used in inverse reinforcement learning to compute a value function with basis reward functions [Ng and Russell, 2000]. The generalization to options is as follows. First we introduce the *discounted state occupancy function*, $u^o$, of option $o\langle \pi, \beta \rangle$:

$$
u^o(s, s') = \mathbb{E}_{s,o}\Big[\, \sum_{k=0}^{T-1} \gamma^k \, \mathbb{I}_{\{s_k = s'\}} \,\Big].
\tag{2}
$$

Then,

$$
R^o(s) = \sum_{s' \in \mathbb{S}} r^\pi(s') \, u^o(s, s'),
$$

where $r^\pi$ is the expected immediate reward vector under $\pi$ and $\langle \mathcal{R}^a \rangle$, *i.e.,* for any $s \in \mathbb{S}$, $r^\pi(s) = \mathbb{E}_{s,\pi}[r_1]$. For convenience, we shall also treat $u^o(s, \cdot)$ as a vector and write $u^o(s)$ to denote it as a vector. To clarify the independence of $u^o$ from the reward function, it is helpful to first note that every MDP can be viewed as the combination of an immediate reward function, $\langle \mathcal{R}^a \rangle$, and a *reward-less MDP*, $\mathcal{M} = \langle \gamma, \mathbb{S}, \mathbb{A}, \langle \mathcal{P}^a \rangle \rangle$.

**Definition 1** *The UOM of option $o$ in a reward-less MDP is defined by $\langle u^o, p^o \rangle$, where $u^o$ is the option's discounted state occupancy function, defined by* (2)*, and $p^o$ is the option's discounted terminal state distribution, defined by* (1)*.*

The main result of this section is the following theorem. All the proofs of the theorems in this paper can be found in an extended paper.

**Theorem 1** *Fix an option $o = o\langle \pi, \beta \rangle$ in a reward-less MDP $\mathcal{M}$, and let $u^o$ be the occupancy function underlying $o$ in $\mathcal{M}$. Let $\langle \mathcal{R}^a \rangle$ be some immediate reward function. Then, for any state $s \in \mathbb{S}$, the return of option $o$ with respect to $\mathcal{M}$ and $\langle \mathcal{R}^a \rangle$ is given by by $R^o(s) = (u^o(s))^\top r^\pi$.*

## 4  UOMs with Linear Function Approximation

In this section, we introduce linear universal option models which use linear function approximation to compactly represent reward independent option-models over a potentially large state space. In particular, we build upon previous work where the approximate solution has been obtained by solving the so-called projected Bellman equations.     We assume that we are given a function $\phi : \mathbb{S} \to \mathbb{R}^n$, which maps any state $s \in \mathbb{S}$ into its $n$-dimensional *feature representation* $\phi(s)$. Let $V_\theta : \mathbb{S} \to \mathbb{R}$ be defined by $V_\theta(s) = \theta^\top \phi(s)$, where the vector $\theta$ is a so-called weight-vector.[2] Fix an initial distribution $\mu$ over the states and an option $o = o\langle \pi, \beta \rangle$. Given a reward function $\mathcal{R} = \langle \mathcal{R}^a \rangle$, the TD(0) approximation $V_{\theta(\mathrm{TD}, \mathcal{R})}$ to $R^o$ is defined as the solution to the following projected Bellman equations [Sutton and Barto, 1998]:

$$\mathbb{E}_{\mu,o}\Big[ \sum_{k=0}^{T-1} \{ r_{k+1} + \gamma V_\theta(s_{k+1}) - V_\theta(s_k) \} \, \phi(s_k) \Big] = 0. \tag{3}$$

Here $s_0$ is sampled from $\mu$, the random variables $(r_1, s_1, r_2, s_2, \ldots)$ and $T$ (the termination time) are obtained by following $o$ from this initial state until termination. It is easy to see that if $\gamma = 0$ then $V_{\theta(\mathrm{TD}, \mathcal{R})}$ becomes the *least-squares* approximation $V_{f(\mathrm{LS}, \mathcal{R})}$ to the immediate rewards $\mathcal{R}$ under $o$ given the features $\phi$. The least-squares approximation to $\mathcal{R}$ is given by $f^{(\mathrm{LS}, \mathcal{R})} = \arg\min_f J(f) = \mathbb{E}_{\mu,o}\Big[ \sum_{k=0}^{T-1} \{ r_{k+1} - f^\top \phi(s_k) \}^2 \Big]$. We restrict our attention to this TD(0) solution in this paper, and refer to $f$ as an (approximate) immediate reward model.

The *TD(0)-based linear UOM* (in short, linear UOM) underlying $o$ (and $\mu$) is a pair of $n \times n$ matrices $(U^o, M^o)$, which generalize the tabular model $(u^o, p^o)$. Given the same sequence as used in defining the approximation to $R^o$ (equation 3), $U^o$ is the solution to the following system of linear equations:

$$\mathbb{E}_{\mu,o}\left[ \sum_{k=0}^{T-1} \{ \phi(s_k) + \gamma U^o \phi(s_{k+1}) - U^o \phi(s_k) \} \phi(s_k)^\top \right] = 0.$$

Let $(U^o)^\top = [u_1, \ldots, u_n]$, $u_i \in \mathbb{R}^n$. If we introduce an artificial "reward" function, $\breve{r}^i = \phi_i$, which is the $i$th feature, then $u_i$ is the weight vector such that $V_{u_i}$ is the TD(0)-approximation to the return of $o$ for the artificial reward function. Note that if we use tabular representation, then $u_{i,s} = u^o(s,i)$ holds for all $s, i \in \mathbb{S}$. Therefore our extension to linear function approximation is backward consistent with the UOM definition in the tabular case. However, this alone would not be a satisfactory justification of this choice of linear UOMs. The following theorem shows that just like the UOMs of the previous section, the $U^o$ matrix allows the separation of the reward from the option models without losing information.

**Theorem 2**  *Fix an option $o = o\langle \pi, \beta \rangle$ in a reward-less MDP, $\mathcal{M} = \langle \gamma, \mathbb{S}, \mathbb{A}, \langle \mathcal{P}^a \rangle \rangle$, an initial state distribution $\mu$ over the states $\mathbb{S}$, and a function $\phi : \mathbb{S} \to \mathbb{R}^n$. Let $U$ be the linear UOM of $o$ w.r.t. $\phi$ and $\mu$. Pick some reward function $\mathcal{R}$ and let $V_{\theta(\mathrm{TD}, \mathcal{R})}$ be the TD(0) approximation to the return $R^o$. Then, for any $s \in \mathbb{S}$,*

$$V_{\theta(\mathrm{TD}, \mathcal{R})}(s) = \left( f^{(\mathrm{LS}, \mathcal{R})} \right)^\top \left( U \phi(s) \right).$$

The significance of this result is that it shows that to compute the TD approximation of an option return corresponding to a reward function $\mathcal{R}$, it suffices to find $f^{(\mathrm{LS}, \mathcal{R})}$ (the least squares approximation of the expected one-step reward under the option and the reward function $\mathcal{R}$), provided one is given the $U$ matrix of the option. We expect that finding a least-squares approximation (solving a regression problem) is easier than solving a TD fixed-point equation. Note that the result also holds for standard policies, but we do not explore this direction in this paper.

*The definition of $M^o$.*  The matrix $M^o$ serves as a state predictor, and we call $M^o$ the *transient matrix* associated with option $o$. Given a feature vector $\phi$, $M^o \phi$ predicts the (discounted) expected feature vector where the option stops. When option $o$ is started from state $s$ and stopped at state $s_T$ in $T$ time steps, we update an estimate of $M^o$ by

$$M^o \leftarrow M^o + \eta(\gamma^T \phi(s_T) - M^o \phi(s))\phi(s)^\top.$$

Formally, $M^o$ is the solution to the associated linear system,

$$\mathbb{E}_{\mu,o}\big[\, \gamma^T \phi(s_T)\phi(s)^\top \,\big] = M^o \, \mathbb{E}_{\mu,o}\big[\, \phi(s)\phi(s)^\top \,\big]. \tag{4}$$

Notice that $M^o$ is thus just the least-squares solution of the problem when $\gamma^T \phi(s_T)$ is regressed on $\phi(s)$, given that we know that option $o$ is executed. Again, this way we obtain the terminal distribution of option $o$ in the tabular case.

A high-level policy $h$ defines a Markov chain over $\mathbb{S} \times \mathcal{O}$. Assume that this Markov chain has a unique stationary distribution, $\mu_h$. Let $(s, o) \sim \mu_h$ be a draw from this stationary distribution. Our goal is to find an option model that can be used to compute a TD approximation to the value function of a high-level policy $h$ (flattened) over a set of options $\mathcal{O}$. The following theorem shows that the value function of $h$ can be computed from option returns and transient matrices.

**Theorem 3** *Let $V_\theta(s) = \phi(s)^\top \theta$. Under the above conditions, if $\theta$ solves*

$$\mathbb{E}_{\mu_h}\big[\, (R^o(s) + (M^o\phi(s))^\top\theta - \phi(s)^\top\theta)\phi(s) \,\big] = 0 \tag{5}$$

*then $V_\theta$ is the* TD(0) *approximation to the value function of $h$.*

Recall that Theorem 2 states that the $U$ matrices can be used to compute the option returns given an arbitrary reward function. Thus given a reward function, the $U$ and $M$ matrices are all that one would need to solve the TD solution of the high-level policy. The merit of $U$ and $M$ is that they are reward independent. Once they are learned, they can be saved and used for different reward functions for different situations at different times.

# 5  Learning and Planning with UOMs

In this section we give incremental, TD-style algorithms for learning and planning with linear UOMs. We start by describing the learning of UOMs while following some high-level policy $h$, and then describe a Dyna-like algorithm that estimates the value function of $h$ with learned UOMs and an immediate reward model.

## 5.1  Learning Linear UOMs

Assume that we are following a high-level policy $h$ over a set of options $\mathcal{O}$, and that we want to estimate linear UOMs for the options in $\mathcal{O}$. Let the trajectory generated by following this high-level policy be $\ldots, s_k, q_k, o_k, a_k, s_{k+1}, q_{k+1}, \ldots$. Here, $q_k = 1$ is the indicator for the event that option $o_{k-1}$ is terminated at state $s_k$ and so $o_k \sim h(s_k, \cdot)$. Also, when $q_k = 0$, $o_k = o_{k-1}$. Upon the transition from $s_k$ to $s_{k+1}, q_{k+1}$, the matrix $U^{o_k}$ is updated as follows:

$$U_{k+1}^{o_k} = U_k^{o_k} + \eta_k^{o_k}\, \delta_{k+1}\, \phi(s_k)^\top, \quad \text{where}$$
$$\delta_{k+1} = \phi(s_k) + \gamma U_k^{o_k}\phi(s_{k+1})\mathbb{I}_{\{q_{k+1}=0\}} - U_k^{o_k}\phi(s_k),$$

and $\eta_k^{o_k} \ge 0$ is the learning-rate at time $k$ associated with option $o_k$. Note that when option $o_k$ is terminated the temporal difference $\delta_{k+1}$ is modified so that the next predicted value is zero.

The $\langle M^o \rangle$ matrices are updated using the least-mean square algorithm. In particular, matrix $M^{o_k}$ is updated when option $o_k$ is terminated at time $k + 1$, *i.e.,* when $q_{k+1} = 1$. In the update we need the feature ($\tilde{\phi}_\cdot$) of the state which was visited at the time option $o_k$ was selected and also the time elapsed since this time ($\tau_\cdot$):

$$M_{k+1}^{o_k} = M_k^{o_k} + \tilde{\eta}_k^{o_k}\mathbb{I}_{\{q_{k+1}=1\}}\left\{\gamma^{\tau_k}\phi(s_{k+1}) - M_k^{o_k}\tilde{\phi}_k\right\}\tilde{\phi}_k^\top,$$
$$\tilde{\phi}_{k+1} = \mathbb{I}_{\{q_{k+1}=0\}}\tilde{\phi}_k + \mathbb{I}_{\{q_{k+1}=1\}}\phi(s_{k+1}),$$
$$\tau_{k+1} = \mathbb{I}_{\{q_{k+1}=0\}}\tau_k + 1.$$

These variables are initialized to $\tau_0 = 0$ and $\tilde{\phi}_0 = \phi(s_0)$.

The following theorem states the convergence of the algorithm.

**Theorem 4** *Assume that the stationary distribution of $h$ is unique, all options in $\mathcal{O}$ terminate with probability one and that all options in $\mathcal{O}$ are selected at some state with positive probability.[3] If the step-sizes of the options are decreased towards zero so that the Robbins-Monro conditions hold for them, i.e., $\sum_{i(k)} \eta_{i(k)}^o = \infty$, $\sum_{i(k)} (\eta_{i(k)}^o)^2 < \infty$, and $\sum_{j(k)} \tilde{\eta}_{j(k)}^o = \infty$, $\sum_{j(k)} (\tilde{\eta}_{j(k)}^o)^2 < \infty$,[4] then for any $o \in \mathcal{O}$, $M_k^o \to M^o$ and $U_k^o \to U^o$ with probability one, where $(U^o, M^o)$ are defined in the previous section.*

## 5.2 Learning Reward Models

In conventional settings, a single reward signal will be contained in the trajectory when following the high level policy, $\ldots, s_k, q_k, o_k, a_k, r_{k+1}, s_{k+1}, q_{k+1}, \ldots$. We can learn for each option an immediate reward model for this reward signal. For example, $f^{o_k}$ is updated using least mean squares rule:

$$f_{k+1}^{o_k} = f_k^{o_k} + \tilde{\eta}_k^{o_k} \mathbb{I}_{\{q_{k+1}=0\}} \left\{ r_{k+1} - f^{o_k \top} \phi(s_k) \right\} \phi(s_k).$$

In other settings, immediate reward models can be constructed in different ways. For example, more than one reward signal can be of interest, so multiple immediate reward models can be learned in parallel. Moreover, such additional reward signals might be provided at any time. In some settings, an immediate reward model for a reward function can be provided directly from knowledge of the environment and features where the immediate reward model is independent of the option.

## 5.3 Policy Evaluation with UOMs and Reward Models

Consider the process of policy evaluation for a high-level policy over options from a given set of UOMs when learning a reward model. When starting from a state $s$ with feature vector $\phi(s)$ and following option $o$, the return $R^o(s)$ is estimated from the reward model $f^o$ and the expected feature occupancy matrix $U^o$ by $R^o(s) \approx (f^o)^\top U^o \phi(s)$. The TD(0) approximation to the value function of a high-level policy $h$ can then be estimated online from Theorem 3. Interleaving updates of the reward model learning with these planning steps for $h$ gives a Dyna-like algorithm.

# 6 Empirical Results

In this section, we provide empirical results on choosing game units to execute specific policies in a simplified real-time strategy game and recommending articles in a large academic database with more than one million articles. We compare the UOM method with a method of Sorg and Singh (2010), who introduced the *linear-option expectation model (LOEM)* that is applicable for evaluating a high-level policy over options. Their method estimates $(M^o, b^o)$ from experience, where $b^o$ is equal to $(U^o)^\top f^o$ in our formulation. This term $b^o$ is the expected return from following the option, and can be computed incrementally from experience once a reward signal or an immediate reward model are available.

*A simplified Star Craft 2 mission.* We examined the use of the UOMs and LOEMs for policy evaluation in a simplified variant of the real-time strategy game Star Craft 2, where the task for the player was to select the best game unit to move to a particular goal location. We assume that the player has access to a black-box game simulator. There are four game units with the same constant dynamics. The internal status of these units dynamically changes during the game and this affects the reward they receive in enemy controlled territory. We evaluated these units, when their rewards are as listed in the table below (the rewards are associated with the previous state and are not action-contingent). A game map is shown in Figure 1 (a). The four actions could move a unit left, right, up, or down. With probability $2/3$, the action moved the unit one grid in the intended direction. With probability $1/3$, the action failed, and the agent was moved in a random direction chosen uniformly from the other three directions. If an action would move a unit into the boundary, it remained in the original location (with probability one). The discount factor was $0.9$. Features were a lookup table over the $11 \times 11$ grid. For all algorithms, only one step of planning was applied per action selection. The

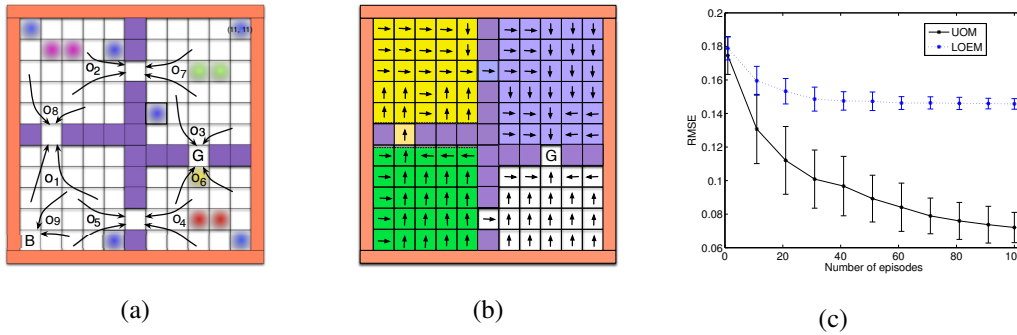

Figure 1: (a) A Star Craft local mission map, consisting of four bridged regions, and nine options for the mission. (b) A high-level policy $h = <o_1, o_2, o_3, o_6>$ initiates the options in the regions, with deterministic policies in the regions as given by the arrows: $o_1$ (green), $o_2$ (yellow), $o_3$ (purple), and $o_6$ (white). Outside these regions, the policies select actions uniformly at random. (c) The expected performance of different units can be learned by simulating trajectories (with the standard deviation shown by the bars), and the UOM method reduces the error faster than the LOEM method.

planning step-size for each algorithm was chosen from $0.001, 0.01, 0.1, 1.0$. Only the best one was reported for an algorithm. All data reported were averaged over 30 runs.

We defined a set of nine options and their corresponding policies, shown in Figure 1 (a), (b). These options are specified by the locations where they terminate, and the policies. The termination location is the square pointed to by each option's arrows.

|  | Game Units | | | |
|---|---|---|---|---|
| Enemy Locations | *Battlecruiser* | *Reapers* | *Thor* | *SCV* |
| fortress (yellow) | 0.3 | -1.0 | 1.0 | -1.0 |
| ground forces (green) | 1.0 | 0.3 | 1.0 | -1.0 |
| viking (red) | -1.0 | -1.0 | 1.0 | -1.0 |
| cobra (pink) | 1.0 | 0.5 | -1.0 | -1.0 |
| minerals (blue) | 0 | 0 | 0 | 1.0 |

Four of these are "bridges" between regions, and one is the position labeled "B" (which is the player's base at position $(1,1)$). Each of the options could be initiated from anywhere in the region in which the policy was defined. The policies for these options were defined by a shortest path traversal from the initial location to the terminal location, as shown in the figure. These policies were not optimized for the reward functions of the game units or the enemy locations.

To choose among units for a mission in real time, a player must be able to efficiently evaluate many options for many units, compute the value functions of the various high-level policies, and select the best unit for a particular high-level goal. A high-level policy for dispatching the game units is defined by initiating different options from different states. For example, a policy for moving units from the base "B" to position "G" can be, $h = <o_1, o_2, o_3>$. Another high-level policy could move another unit from upper left terrain to "G" by a different route with $h' = <o_8, o_5, o_6>$.

We evaluated policy $h$ for the Reaper unit above using UOMs and LOEMs. We first pre-learned the $U^o$ and $M^o$ models using the experience from 3000 trajectories. Using a reward function that is described in the above table, we then learned $f^o$ for the UOM and and $b^o$ for the LEOM over 100 simulated trajectories, and concurrently learned $\theta$. As shown in Figure 1(c), the UOM model learns a more accurate estimate of the value function from fewer episodes, when the best performance is taken across the planning step size. Learning $f^o$ is easier than learning $b^o$ because the stochastic dynamics of the environment is factored out through the pre-learned $U^o$. These constructed value functions can be used to select the best game unit for the task of moving to the goal location.

This approach is computationally efficient for multiple units. We compared the computation time of LOEMs and UOMs with linear Dyna on a modern PC with an Intel $1.7$GHz processor and $8$GB RAM in a MATLAB implementation. Learning $U^o$ took $81$ seconds. We used a recursive least-squares update to learn $M^o$, which took $9.1$ seconds. Thus, learning an LOEM model is faster than learning a UOM for a single fixed reward function, but the UOM can produce an accurate option return quickly for each new reward function. Learning the value function incrementally from the 100

trajectories took $0.44$ seconds for the UOM and $0.61$ seconds for the LOEM. The UOM is slightly more efficient as $f^o$ is more sparse than $b^o$, but it is substantially more accurate, as shown in Figure 1(c). We evaluated all the units and the results are similar.

*Article recommendation.* Recommending relevant articles for a given user query can be thought of as predicting an expected return of an option for a dynamically specified reward model. Ranking an article as a function of the links between articles in the database has proven to be a successful approach to article recommendation, with PageRank and other link analysis algorithms using a random surfer model [Page et al., 1998]. We build on this idea, by mapping a user query to a reward model and pre-specified option for how a reader might transition between articles. The ranking of an article is then the expected return from following references in articles according to the option. Consider the policy of performing a random-walk between articles in a database by following a reference from an article that is selected uniformly at random. An article receives a positive reward if it matches a user query (and is otherwise zero), and the value of the article is the expected discounted return from following the random-walk policy over articles. More focused reader policies can be specified as following references from an article with a common author or keyword.

We experimented with a collection from DBLP that has about $1.5$ million articles, 1 million authors, and 2 millions citations [Tang et al., 2008]. We assume that a user query $q$ is mapped directly to an option $o$ and an immediate reward model $f_q^o$. For simplicity in our experiment, the reward models are all binary, with three non-zero features drawn uniformly at random. In total we used about $58$ features, and the discount factor was 0.9. There were three policies. The first followed a reference selected uniformly at random, the second selected a reference written by an author of the current article (selected at random), and the third selected a reference with a keyword in common with the current article. Three options were defined from these policies, where the termination probability beta was 1.0 if no suitable outgoing reference was available and 0.25 otherwise. High-level policies of different option sequences could also be applied, but were not tested here. We used bibliometric features for the articles extracted from the author, title, venue fields.

We generated queries $q$ at random, where each query specified an associated option $o$ and an option-independent immediate reward model $f_q^o = f_q$. We then computed their value functions. The immediate reward model is naturally constructed for these problems, as the reward comes from the starting article based on its features, so it is not dependent on the action taken (and thus not the option). This approach is appropriate in article recommendation as a query can provide both terms for relevant features (such as the venue), and how the reader intends to follow references in the paper. For the UOM based approach we pre-learned $U^o$, and then computed $U^o f_q^o$ for each query. For the LOEM approach, we learned a $b_q$ for each query by simulating 3000 trajectories in the database (the simulated trajectories were shared for all the queries). The computation time (in seconds) for the UOM and LOEM approaches are shown in the table below, which shows that UOMS are much more computationally efficient than LOEM.

| Number of reward functions | 10 | 100 | 500 | 1,000 | 10,000 |
|---|---|---|---|---|---|
| LOEM | 0.03 | 0.09 | 0.47 | 0.86 | 9.65 |
| UOM | 0.01 | 0.04 | 0.07 | 0.12 | 1.21 |

## 7 Conclusion

We proposed a new way of modelling options in both tabular representation and linear function approximation, called the universal option model. We showed how to learn UOMs and how to use them to construct the TD solution of option returns and value functions of policies, and prove their theoretical guarantees. UOMs are advantageous in large online systems. Estimating the return of an option given a new reward function with the UOM of the option is reduced to a one-step regression. Computing option returns dependent on many reward functions in large online games and search systems using UOMs is much faster than using previous methods for learning option models.

## Acknowledgment

Thank the reviewers for their comments. This work was supported by grants from Alberta Innovates Technology Futures, NSERC, and Department of Science and Technology, Government of India.

## Footnotes

[1] Here, $s_{k+1}$ is sampled from $\mathcal{P}^{a_k}(s_k, \cdot)$ and the mean of $r_{k+1}$ is $\mathcal{R}^{a_k}(s_k, s_{k+1})$.

[2] Note that the subscript in $V$. always means the TD weight vector throughout this paper.

[3]Otherwise, we can drop the options in $\mathcal{O}$ which are never selected by $h$.

[4] The index $i(k)$ is advanced for $\eta_{i(k)}^o$ when following option $o$, and the index $j(k)$ is advanced for $\tilde{\eta}_{j(k)}^o$ when $o$ is terminated. Note that in the algorithm, we simply wrote as $\eta_{i(k)}^o$ as $\eta_k^o$ and $\tilde{\eta}_{j(k)}^o$ as $\tilde{\eta}_k^o$.

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
