[Reviews · NeurIPS 2014]

Submitted by Assigned_Reviewer_22

This paper introduces a framework for learning from options in reinforcement learning. An option is a policy which has some probability of terminating at a certain state. This paper introduces the notion of an “option policy”, which is like a high-level policy that allows for multi-step transition between states. They show how to make the option model universal with respect to rewards, and provide an TD-style algorithm for learning with such models.

The universal option model proposed by this paper seems extremely interesting and useful, and I like the cool theory that they provide. It appears to me that this option model is a very elegant way of getting around the "memoryless" property of MDPs while still retaining much of their simplicity. I am not an expert on reinforcement learning, and hence I cannot judge the novelty of this work. However, I like the model, and as a result, I would recommend acceptance.
Summary: This paper introduces an interesting model of reinforcement learning using options. I cannot judge the novelty of the work, but I like the model and hence will recommend acceptance.

Submitted by Assigned_Reviewer_46

This paper proposes and analyses "Universal Option Models", an extension of discounted occupancy functions (Ng & Russell, 2000) to "options" (Sutton et al., 1999) – high-level actions that consist of a policy and a state-dependent termination probability function. After establishing a connection between an options discounted state occupancy function and an option rewards (Theorem 1) the application of options to learning and planning in large state spaces via linear function approximation is considered. Consistency and convergence results are given (Theorems 2, 3, and 4) before the efficacy of the technique relative to an earlier option-based technique is demonstrated on two domains.

In general, the paper is clearly motivated and the main ideas are presented in a logical fashion. The experimental methodology is clearly explained and the results show a clear and significant improvement over the existing method by Sorg & Singh (2010). I am admittedly not well-versed in the reinforcement learning literature but this seems to be a strong point of the paper. However, I am surprised at the lack of related reinforcement learning work between 2000 and 2010. The references only cite work on PageRank (2002) and citation networks (2008) in the intervening years.

My main concerns surround the theoretical results in the paper. The notion of a "reward-less MDP" appears in Theorems 1 & 2 but is not formally defined anywhere. A theorem that rests on a definition in "scare quotes" makes me nervous as it is not entirely clear what the theorem applies to. It would be good to do so and provide a brief discussion of what is being captured by this definition.

The use of a inner product between the vectors $u^o(s)$ and $r^\pi$ at line 151 is very confusing. Just before Definition 1, $u^o(s)$ is defined to be a vector indexed by states over the second argument of $u^o$ and $r^\pi$ is also a vector indexed by states. Accordingly, my interpretation of the inner product $(u^o(s))^\top r^\pi$ would be $\sum_{s'} u^o(s, s') r^\pi(s')$ which is *not* the same as the sum in (4) for $R^o(s)$. No part of the proof of Theorem 1 in the appendix does not use the inner product notation but instead shows the return of option $o$ satisfies (4) directly. As far as I can tell, the vector form of $R^o$ is not relied upon later in the paper so I do not believe this has serious implications for the rest of the paper.

Theorem 4 refers to the "Robbins-Monro conditions" but, as far as I can tell these are not given or referenced anywhere in the paper. Even if these are common knowledge in RL it is poor practice to be this imprecise when stating theorems. Furthermore, the "proof" of this result is really only a sketch. Given that there is no page limit for appendices a full proof should be provided.
Summary: While the main idea, analysis, and experimental result seem novel, interesting, and significant, some of the theoretical results are poorly expressed and the proof of Theorem 4 is not at an appropriate level of rigour.

Submitted by Assigned_Reviewer_47

The paper describes a new technique for handling "options" (abstract, high-level actions) in reinforcement learning. The paper argues that the new approach is more efficient, and has the advantage of producing models that are independent of the reward function. In addtion, it is shown that the approach can be extended to linear function approximation. Experiments are conducted in two domains.

These results appear to be a significant advance in the handling of options, which seem to be a natural and potentially powerful approach for dealing with complex environments. The paper is very clearly written. The research is solid and high quality.

I would suggest that the authors also provide a complexity analysis of their approach with regard to its time and space requirements. The paper would also benefit from further discussion of any disadvantages of the approach, or situations in which it might not be appropriate.
Summary: Significant results, very clearly written.
Author Feedback
Author rebuttal: We thank the reviewers for their comments. Below, we clarify the text
with respect to the issues they raised.

reward-less MDPs: ``The notion of a "reward-less MDP" appears in
Theorems 1 and 2 but is not formally defined anywhere''.

The notion of reward-less MDPs is used to clarify the independence of
$u^o$ (defined in equation 3) from the immediate reward function. It
is helpful to note that every MDP can be viewed as the combination of
an immediate reward function, $< R^a > $, and a reward-less MDP, $\gamma,
S, A, < P^a > $. We have added this clarification right before Definition
1.

We had a typo in the definition of $R^o$ in equation 4 where
$r^{\pi}(s)$ should have been $r^{\pi}(s')$.

Theorem 4 has been revised to provide more detail.